# Effects of Silica Shell Encapsulated Nanocrystals on Active χ-Fe_5_C_2_ Phase and Fischer–Tropsch Synthesis

**DOI:** 10.3390/nano12203704

**Published:** 2022-10-21

**Authors:** Seunghee Cha, Heewon Kim, Hyunkyung Choi, Chul Sung Kim, Kyoung-Su Ha

**Affiliations:** 1Department of Chemical and Biomolecular Engineering, Sogang University, 35 Baekbeom-ro, Mapo-gu, Seoul 04107, Korea; 2Department of Physics, Kookmin University, 77 Jeongneung-ro, Seongbuk-gu, Seoul 02707, Korea

**Keywords:** Fischer–Tropsch, iron carbide, encapsulation, ordered mesoporous silica

## Abstract

Among various iron carbide phases, χ-Fe_5_C_2_, a highly active phase in Fischer–Tropsch synthesis, was directly synthesized using a wet-chemical route, which makes a pre-activation step unnecessary. In addition, χ-Fe_5_C_2_ nanoparticles were encapsulated with mesoporous silica for protection from deactivation. Further structural analysis showed that the protective silica shell had a partially ordered mesoporous structure with a short range. According to the XRD result, the sintering of χ-Fe_5_C_2_ crystals did not seem to be significant, which was believed to be the beneficial effect of the protective shell providing restrictive geometrical space for nanoparticles. More interestingly, the protective silica shell was also found to be effective in maintaining the phase of χ-Fe_5_C_2_ against re-oxidation and transformation to other iron carbide phases. Fischer–Tropsch activity of χ-Fe_5_C_2_ in this study was comparable to or higher than those from previous reports. In addition, CO_2_ selectivity was found to be very low after stabilization.

## 1. Introduction

Fischer–Tropsch synthesis (FTS) is known to be one of the most important reactions in gas-to-liquid processes since it produces synthetic oils and basic chemicals such as naphtha, gasoline, diesel, waxy products, ethylene, and propylene [1,2,3]. Among the FTS catalysts, iron-based catalysts are attracting more attraction due to their low cost and flexibility of operation to yield various kinds of products [4,5,6]. There have been many research articles dealing with iron carbide phases as the active phases of Fischer–Tropsch synthesis. χ-Fe_5_C_2_ is usually considered to be one of the most active phases [4,7]. However, the χ-Fe_5_C_2_ phase has been known susceptible to significant phase transformation to other carbides or oxide phases during reaction [8,9]. Despite the effort to protect and decrease the changing rate in phases, a K-promoted Fe catalyst showed a transition from the χ-Fe_5_C_2_ phase into the ε’-Fe_2.2_C phase [10], which is also known as an active phase for FTS. It was reported that the growth of an amorphous carbon layer formed around the catalyst particle was attributed to the resistance against re-oxidation and had an influence on the phase change between active iron carbide phases. Despite the beneficial feature of resistance against re-oxidation, the gradual decrease in conversion was not avoided, and the original phase did not seem to be well protected either.

Regarding the formation degree of iron carbides when using syngas as an activation gas [11,12,13], it was reported that a decrease in H_2_/CO ratio during the pre-activation process could facilitate carburization to give more active iron carbides [14]. According to Smit et al. [15], thermodynamics with chemical potential could explain the formations of χ-Fe_5_C_2_, θ-Fe_3_C, Fe_7_C_3_, and ϵ-carbides and their relative stabilities. They also argued that the catalyst composed mainly of the χ-Fe_5_C_2_ phase prepared under high chemical potential was found to be quite vulnerable to oxidation during FTS because the amorphous iron carbides in the catalyst appeared to be oxidized to iron oxide crystals while the intensity of the χ-Fe_5_C_2_ phase in the XRD spectrum remained unchanged. As far as the formation of the χ-Fe_5_C_2_ phase is concerned, a wet-chemical synthesis was introduced [8], and it produced a pure χ-Fe_5_C_2_ phase only, by forming octadecylamine and a Fe carbonyl complex followed by carburization with C_2_–C_3_ molecules from thermally fragmented octadecylamine at around 350 °C, and the procedure and process of synthesis were reconstructed in a later study [16]. These authors argued that the phase change of χ-Fe_5_C_2_ phase to iron oxide was ascribed to the cause of deactivation instead of hydrocarbon shielding during FTS. Tang et al. reported that χ-Fe_5_C_2_ phase nanoparticles synthesized by a wet-chemical route might be more vulnerable to re-oxidation during a reaction than iron carbides prepared by gas pre-treatment [17]. Regarding the χ-Fe_5_C_2_ phase, a highly active carbide phase was reported to be prepared by infiltration of the iron precursor melt into graphene flakes. The resulting catalyst was found to have a highly active χ-Fe_5_C_2_ phase in terms of total CO conversion at high space velocity, whereas CO to CO_2_ conversion was also very high and comparable to CO to hydrocarbon conversion [18]. To protect against sintering, core–shell-like catalyst particles were developed and Co_3_O_4_@SiO_2_ was prepared and tested for FTS [19]. It was reported that enlarging the pore size of the shell could increase the dispersion of active core particles and catalytic performance as well. In addition, the encapsulation improved catalytic stability. 

In this study, χ-Fe_5_C_2_ phase nanoparticles were prepared to elucidate the performance of pure χ-Fe_5_C_2_ instead of a mixture of iron carbides. By encapsulating the nanoparticle with a partially ordered mesoporous silica shell, the effects of the protective shell on sintering and re-oxidation of χ-Fe_5_C_2_ were investigated by XRD, TEM with SAED, physisorption, and other methods. In addition, the catalytic performance was tested and compared with the performances of catalysts having different and mixed iron carbides. 

## 2. Materials and Methods

### 2.1. Materials

The octadecylamine (85% purity), iron pentacarbonyl (Fe(CO)_5_, 99.99% purity), polyvinylpyrrolidone (PVP, average MW = 40,000), triethanolamine (TEA, 98% purity), and tetraethyl orthosilicate (TEOS, 98% purity) were purchased from Sigma Aldrich, Burlington, MA, USA. The hexadecyltrimethylammonium bromide (CTAB) was purchased from TCI Chemicals, Tokyo, Japan. All chemicals were used without further purification.

### 2.2. Catalyst Preparation 

#### 2.2.1. Synthesis of χ-Fe_5_C_2_ Nanoparticles

Iron carbide nanoparticle preparation was conducted as follows: The χ-Fe_5_C_2_ nanoparticles were synthesized via a wet-chemical route [16]. First, 30 g of octadecylamine and 0.226 g of CTAB were introduced into a 3-neck round bottom flask and degassed with inert gas flow at room temperature. The synthesis of iron carbide nanoparticles proceeded under inert conditions during the whole procedure. Consistent N_2_ flow was connected to the mixture during the interval of the synthesis of iron carbide nanoparticles. Then the mixture was heated to 120 °C. When octadecylamine and CTAB were fully mixed, 1 mL of iron pentacarbonyl (Fe(CO)_5_) was added into the mixture, which was heated to 180 °C and maintained for 30 min, followed by being heated to 350 °C and maintained for 5 min. Thereafter, the mixture was cooled to room temperature, and passivation was conducted for 15 min using a 5% O_2_/N_2_ mixture flow. The mixture was then dispersed in hexane and washed several times. During the wash, magnets were used for the separation of the χ-Fe_5_C_2_ nanoparticles having magnetic properties. The washing temperature was room temperature, and about 1 min of washing with hexane was repeated around 20 times until the separation of iron carbide nanoparticles. About 250 mL of hexane was needed for the repetitive separation process. 

#### 2.2.2. Synthesis of χ-Fe_5_C_2_@SiO_2_ Catalysts

First, 0.07 g of χ-Fe_5_C_2_ nanoparticles was mixed with 3 g of polyvinylpyrrolidone (PVP) in 50 mL of ethanol to meet the final weight percent, and PVP-protected χ-Fe_5_C_2_ nanoparticles were dispersed in ethanol by sonication. This mixture was named solution A. Encapsulation of iron carbide nanoparticles with silica was conducted as follows: First, 8.4 g of CTAB and 72 g of deionized water were mixed at 60 °C for 30 min using a rotary mixer, and solution A was added dropwise and mixed. This mixture was mixed at 60 °C for at least 30 min and named solution B. At the same time, solution C was prepared by mixing 6 g of tetraethyl orthosilicate (TEOS) and 45 g of triethanolamine (TEA) in a Teflon-lined stainless-steel autoclave, and hydrothermal synthesis was conducted at 90 °C for 20 min. Finally, solutions B and C were mixed after they had cooled to room temperature. The final mixture was mixed for 48 h using a rotating vessel. The mixture was then washed out with deionized water and ethanol and dried at 60 °C in a vacuum oven. The resulting samples were named χ-Fe_5_C_2_@SiO_2__L. Through the reconstruction of the synthesis method, χ-Fe_5_C_2_@SiO_2__H, which has higher Fe content than Fe_5_C_2_@SiO_2__L, was synthesized. The amounts of solution B and solution C were halved to increase Fe loading in catalysts. Considering the decreased amounts of solvent and solution, 0.13 g of χ-Fe_5_C_2_ nanoparticles was used in the case of the χ-Fe_5_C_2_@SiO_2__H catalyst. In addition, the contact of PVP-protected χ-Fe_5_C_2_ nanoparticles with the aqueous solution was allowed only until the color of the mixture turned orange. The orange color indicated that the oxidation of χ-Fe_5_C_2_ nanoparticles occurred. χ-Fe_5_C_2_@SiO_2__L (3.5 wt.%) and χ-Fe_5_C_2_@SiO_2__H (12.2 wt.%) catalysts were synthesized in the same way, except the amounts of χ-Fe5C2 nanoparticles and the contact time of the nanoparticles and the aqueous solution were different.

### 2.3. Catalyst Characterization

N_2_-physisorption analysis was conducted using an ASAP 2020 analyzer (Micromeritics, Inc., Norcross, GA, USA) at −196 °C. Prior to the physisorption measurements, the samples were degassed at 150 °C for 12 h under vacuum condition. X-ray diffraction (XRD) analysis was performed using a Rigaku SmartLab with a D/teX Ultra 250 X-ray diffractometer with Cu Kα radiation (λ = 0.154 nm, 40 kV, 40 mA) at 2θ ranging from 10° to 65° for all samples. Small-angle X-ray scattering (SAXS) was conducted on an Empyrean with ScatterX78 PANalytical using 45 kV and 40 mA. A JEM-2100F high-resolution transmission electron microscope was used to obtain TEM images and SAED patterns, and the working voltage was 200 kV. X-ray photoelectron spectroscopy (XPS) was conducted using a K-Alpha X-ray Photoelectron Spectrometer System (Thermo Fisher Scientific Inc., Waltham, Massachusetts, USA). Prior to analysis, fresh and spent catalysts were pressed into thin pellets and etched with low-energy Ar ion irradiation for 400 s. During the experiments, a monochromatic Al Kα (1486.6 eV) was adopted, and approximately 10^−7^ Pa of vacuum condition was preserved. The correction of binding energy (BE) was performed referencing the BE of C 1s (284.6 eV). Inductively coupled plasma optical emission spectroscopy (ICP-OES) was performed using a Thermo Scientific iCAP 7000 to verify the content of iron in the catalyst samples. Thermogravimetric analysis (TGA) was performed with a Q50 instrument (TA Instruments) to identify the change in the silica pore structure of the fresh catalyst. The catalyst was heated under N_2_ flow from 110 °C to 950 °C at a ramping rate of 20 °C/min. Fourier transform infrared (FT-IR) data were collected with a Thermo Nicolet iS50 FT-IR spectrometer (Thermo Fisher Scientific Inc., Waltham, MA, USA) forming a KBr pellet. A Mössbauer spectrometer of the electromechanical type with a ^57^Co source in a Rh matrix was used in the constant-acceleration mode. The spectrometer was calibrated by collecting the Mössbauer spectra of a standard α-Fe foil at room temperature. To produce a uniform thickness over the area of the Mössbauer absorber, each sample was clamped between two beryllium disks of 0.005-inch thickness and 1-inch diameter. The Mössbauer spectra were fitted by the least-squares method and provided the hyperfine field (H_hf_), isomer shift (δ), electric quadrupole splitting (ΔE_Q_), and relative area of Fe.

### 2.4. Catalyst Evaluation

The Fischer–Tropsch activity test of the catalysts was conducted using a laboratory-scaled micro fixed-bed reactor system. For the reaction, 0.3 g of catalyst without dilution was loaded in the reactor. No pre-treatment steps were performed before the reaction, and syngas with a composition of H_2_:CO:CO_2_:Ar = 58.5:27.7:9.1:4.8 was used. The reaction was conducted at a gas hourly space velocity (GHSV) of 4000 mL/g_cat_/h. The reaction was conducted for 24 h, and the temperature and pressure were preserved at 300 °C and 10 bar, respectively. The effluent from the reactor was analyzed using an on-line gas chromatograph (6500GC, Young Lin Instrument Co., Anyang, Korea) equipped with a thermal conductivity detector (TCD) and flame ionization detector (FID). A Carboxen 1000 packed column was used to detect Ar, CO, CH_4_, and CO_2_, and a GS-GASPRO capillary column was used to identify hydrocarbons.

## 3. Results and Discussion

### 3.1. Characterization of χ-Fe_5_C_2_ Nanoparticles 

After the synthesis of χ-Fe_5_C_2_ nanoparticles by the wet-chemical route, the average crystallite size and structure of the nanoparticles were investigated by XRD analysis. In Figure 1a, the X-ray diffraction pattern shows that only the χ-Fe_5_C_2_ phase (JCPDS No. 036-1248) existed in the nanoparticles. The average crystallite size was 20.50 nm, which was calculated using Scherrer’s equation at 2θ = 39.4°, and the peak was known as the (020) plane of χ-Fe_5_C_2_. In addition, the broad peak that appeared in the range of 2θ = 16–33° revealed the existence of amorphous carbon around the particles, as shown in Appendix A. TEM images of nanoparticles and their size distribution are shown in Figure 1b and its inset, respectively. Stochastic treatment indicated that the average particle size was observed as around 44 nm. The sizes of the nanoparticles were much larger than that of the crystallite observed by XRD. This is partly because the nanoparticles had polycrystalline characteristics, as shown in Figure 1c, and a similar result was also reported elsewhere [20]. As shown in Figure 1c, different lattices were observed and identified in one nanoparticle. A more detailed observation of the HRTEM image showed that the marked lattice distances of 0.208 nm and 0.239 nm indicated those of the (021) and (20-2) planes of the χ-Fe_5_C_2_ phase, respectively. In Figure 1d, the iron carbide phase was identified with SAED analysis, and only the χ-Fe_5_C_2_ phase was identified. 

### 3.2. Morphology and Structure of Encapsulated Catalyst Particles χ-Fe_5_C_2_@SiO_2_

After the synthesis of silica-encapsulated catalysts, the structures of the χ-Fe_5_C_2_@SiO_2__L and χ-Fe_5_C_2_@SiO_2__H catalysts were examined and elucidated by XRD analysis. The X-ray diffraction spectrum of χ-Fe_5_C_2_@SiO_2__L showed that the χ-Fe_5_C_2_ phase was not the only phase of iron in the catalyst, and peaks indicating α-Fe_2_O_3_ (JCPDS No. 33-0664) and θ-Fe_3_C (JCPDS No. 35-0772) were found additionally, as shown in Figure 2. The step of encapsulating χ-Fe_5_C_2_ nanoparticles with a silica shell could be ascribed to the oxidation of iron carbide. χ-Fe_5_C_2_ as well as the amorphous iron carbide phase seemed to be oxidized in the first step of the encapsulation process, where nanoparticles were mixed with the aqueous solution at 60 °C for 30 min. Due to the relatively long duration of the first step, the oxidation state of iron species turned out to be α-Fe_2_O_3_ rather than Fe_3_O_4_. According to Tang et al., χ-Fe_5_C_2_ nanoparticles synthesized by a wet-chemical route might be more vulnerable to re-oxidation during the reaction than iron carbides prepared by gas pre-treatment [17]. Therefore, we believed that the oxidation of χ-Fe_5_C_2_ nanoparticles occurred before the completion of the silica shell and oxidation took place in the middle of contact with the aqueous solution at 60 °C. We attempted to encapsulate iron carbide nanoparticles without compromising the active phase following a recipe from the literature [21], but this method proved unsuitable for us because of difficulties in dissolving and incorporating chemical agents such as CTAB and PVP. 

In order to suppress the phase change, the contact time with the aqueous solution mixture was decreased as much as possible when the χ-Fe_5_C_2_@SiO_2__H catalyst was encapsulated with a silica shell. As a result, the XRD spectrum of Fe_5_C_2_@SiO_2__H exhibited the χ-Fe_5_C_2_ (JCPDS No. 36-1248) phase and the Fe_3_O_4_ (JCPDS No. 19-0629) phase, which was a less oxidized phase than the α-Fe_2_O_3_ phase of the χ-Fe_5_C_2_@SiO_2__L catalyst as mentioned above. The number of transformed phases and the degree of oxidation were found to be significantly reduced. 

The morphology of the catalysts was investigated using TEM imaging analysis, and the results are shown in Figure 3. The size of the silica shells was approximately 100 nm in both catalysts, and the iron carbide nanoparticles were encapsulated as a core. The silica shell appeared spherical, and one or two χ-Fe_5_C_2_ nanoparticles appeared to be located inside the shell. In addition, spherical silica structures with diameters much smaller than 100 nm without nanoparticles inside were seen in both catalysts. By comparison between Figure 3b,f, it could be noticed that the χ-Fe_5_C_2_@SiO_2__L catalyst contained more empty silica structures than the χ-Fe_5_C_2_@SiO_2__H catalyst, possibly due to the result of lower iron loading.

The average crystallite size of χ-Fe_5_C_2_ in the χ-Fe_5_C_2_@SiO_2__H catalyst calculated at 2θ = 39.4° from the XRD result was 20.03 nm after the nanoparticles were encapsulated, indicating that the crystal size in the core was almost unchanged. To further verify the structure of the silica shell, N_2_-physisorption analysis was conducted on both catalysts. The N_2_-physisorption results of the χ-Fe_5_C_2_@SiO_2__L catalyst are shown in Appendix A. The BET surface area was 301 m^2^/g, and the average pore size and volume were 7.0 nm and 0.5 cm^3^/g, respectively. The N_2_-physisorption results of the χ-Fe_5_C_2_@SiO_2__H catalyst are shown in Figure 4a,b. The BET surface area calculated from the adsorption isotherm in Figure 4a was 239 m^2^/g. The pore size distribution in Figure 4b shows a relatively sharp peak at ca. 2.3 nm and another broadened peak above 10 nm. The average pore size and pore volume were measured as 5.7 nm and 0.3 cm^3^/g, respectively. The pore volume at approximately 2.3 nm appeared to be small due to residue of PVP and CTAB still remaining after the synthesis. The result of SAXS of the χ-Fe_5_C_2_@SiO_2__H catalyst in Figure 4c shows a slightly broadened peak at approximately 0.13 Å^−1^, and the pore structure could be considered similar to that of MCM-41 since the peak appears very similar to the reflection of the (100) plane of MCM-41 in terms of q vector and profile. The pores of the silica shell appeared to have a wormhole-like and short-range morphology, as shown in TEM images, and it could be corroborated by SAXS results that the pore structure seemed to be a partially ordered mesoporous structure; similar structures can be found in literature elsewhere [19,22,23,24]. The wall thickness of the pore structure was calculated using the d-spacing of the (100) plane and pore size from the N_2_-physisorption result. As the pore structure appeared to be similar to that of MCM-41, the calculation could be simplified by assuming a hexagonal structure. The distance a_0_, the distance between the centers of two adjacent pores, was calculated in the same way as Sibeko et al. proposed [25], and the value was 54.8 Å. As the average pore size considering only mesopores below 10 nm was 2.2 nm, as shown in Figure 4b, the wall thickness was 32.7 Å, which is a slightly higher value than that of MCM-41 from the literature [25] for comparison.

### 3.3. Activity Test Results

Figure 5 shows the catalytic performance of the catalysts, including CO conversion, product selectivity, and CO to CO_2_ and hydrocarbon conversions. As the FTS active phase was directly synthesized, an in situ gaseous pre-treatment was not necessary before the reaction. The catalytic activity of the catalysts was evaluated at 300 °C and 10 bar in a micro fixed-bed reactor. The highest CO conversion over the χ-Fe_5_C_2_@SiO_2__L catalyst was 10.8% at TOS 5.5 h and decreased to 7.4% at the end of the reaction. The low loading amount of the iron species and the phase transformation to α-Fe_2_O_3_ were believed to cause low Fischer–Tropsch synthesis performance. The confining geometry due to the encapsulation with porous silica was believed to play a protective role against deactivation, and the conversion and selectivities of products appeared steady. In addition, the light olefin ratio was higher than any other FTS reaction results in Table 1. The presence of the θ-Fe_3_C phase in an iron-based catalyst is believed to affect the formation of light olefins, as reported in [26]. The generation of CO_2_ was barely seen, whereas the consumption was observed at a very low rate, under 3%. The α-Fe_2_O_3_ phase is known to be the active phase for the reverse water–gas shift (RWGS) reaction [27], and the θ-Fe_3_C phase is also known to be the active phase for CO_2_ hydrogenation [28].

As indicated in Section 3.2, the preparation method was modified so that the Fe loading could be increased and phase transformation could be suppressed as much as possible, as mentioned in Section 2.2.2. In the reaction where the χ-Fe_5_C_2_@SiO_2__H catalyst was used, the CO conversion was greatly enhanced and showed the highest conversion of 38.5%, and the FTS activity reached 44.1 μmol_CO_/g_Fe_/s at TOS 3 h, as shown in Table 1. The degree of FTS activity was further investigated by a comparison with the results in the literature, and the FTS activity of this study appeared to be higher than or comparable to those from the references shown in Table 1. After TOS 3 h, the CO conversion gradually decreased. The overall CO conversion could be attributed to both FTS and water–gas shift (WGS), and the activity of WGS was assessed by CO to CO_2_ conversion, as shown in Figure 5c. During the reaction, CO to CO_2_ conversion consistently decreased, and the final value was measured at a significantly low level of approximately 2%, as shown in Figure 5c. The CO_2_ selectivity was also relatively high in the early stage of the reaction, but decreased to a significantly low level to be 9.9% (not shown). This indicated that WGS activity was high at first but gradually decreased. If WGS activity was disregarded, the CO to hydrocarbon conversion (blue) shown in Figure 5c appeared to be steadier. More specifically, the CO to hydrocarbon conversion seemed greatly stabilized after TOS 10 h and decreased by only about 3.2 percentage points thereafter. It was believed that due to the absence of a calcination step in the catalyst preparation, H_2_O molecules and surface hydroxyl groups associated with the silica shell could have evolved during the reaction, and they seemed to influence WGS activity.

To identify the species and to assess the amounts of released molecules, TG/DTA analysis was performed, and the result is shown in Figure 6a. Two steps of thermal decomposition were shown in the TGA result. The first step was in the temperature range of 150 °C to 360 °C, and this peak agreed with the decomposition peak of CTAB [33]. The second step ranging from 360 °C to 530 °C appeared to be in agreement with the temperature range of thermal decomposition of PVP [34]. The temperature range from 150 °C to 500 °C could be assigned to excess strongly bound water and surface hydroxyl groups [32]. TG/DTA analysis implied that the absence of a calcination step during the preparation of the catalyst caused insufficient removal of the surfactant and water. As mentioned, the evolution of water from the silica shell might have caused excessive WGS activity in the early stage of the reaction.

Figure 6b shows the FT-IR spectra of the χ-Fe_5_C_2_@SiO_2__H catalyst. The broad band at 3000 to 3700 cm^−1^ was assigned to the characteristic bands for water molecules, surface hydroxyls, and hydroxyl functional groups from PVP. Other bands marked in Figure 6b correspond to the characteristic bands of functional groups included in the stabilizing agent, PVP [35]. This corroborated the result of TG/DTA and verified that the evolution of water and hydroxyls could cause WGS in the early stages of the reaction.

### 3.4. Characteristics of Spent Catalyst 

The structure, morphology, and iron species of the spent catalysts were identified and analyzed by XRD, TEM, and Mössbauer emission spectroscopy (MES) methods. Figure 7 shows the XRD patterns of the spent catalysts. The χ-Fe_5_C_2_, θ-Fe_3_C, and α-Fe_2_O_3_ phases were observed in the spent χ-Fe_5_C_2_@SiO_2__L. The kinds of phases were the same, but the amount of χ-Fe_5_C_2_ turned out to be decreased and possibly converted into oxidized phases since the peak at around 50° of χ-Fe_5_C_2_ was barely observed in this spent sample but was originally observed in the fresh catalyst as shown in Figure 2a. Similarly, the spent χ-Fe_5_C_2_ @SiO_2__H was shown to have the same kinds of phases such as χ-Fe_5_C_2_ and Fe_3_O_4_, which were the same phases contained in the fresh one. However, additional but slight oxidation or phase change to Fe_3_O_4_ seemed to occur during the reaction because a new peak at 2θ = 56.9° appeared. 

The Fe 2p XPS spectra of the catalysts are shown in Appendix A. The peaks at binding energies of around 707 eV and 720 eV were allocated to Fe 2p 3/2 and Fe 2p 1/2 of the iron carbide phases, respectively [18]. The peaks situated at about 710 eV and 723 eV were attributed to the main peaks of iron oxide phases of the 2p 3/2 and 2p 1/2 regions, respectively [36,37]. All peaks were positioned at similar binding energies even after the FTS reaction. 

The average crystal size of χ-Fe_5_C_2_ in spent χ-Fe_5_C_2_@SiO_2__L was very difficult to measure since there no unoverlapped peaks of the corresponding phase were observed. The average crystal size of α-Fe_2_O_3_ calculated at 2θ = 35.6° was 23.8 nm, which is very close to that of the fresh catalyst (22.7 nm). Regarding the spent χ-Fe_5_C_2_@SiO_2__H catalyst, the average crystal size of χ-Fe_5_C_2_ could be calculated by the Scherrer equation at 2θ = 39.4°, and the size was 18.75 nm. The crystal size of the Fe_3_O_4_ phase in this spent catalyst was calculated from the (311) diffraction peak using the Scherrer equation, and the crystal size barely changed from 11.85 nm to 12.81 nm. The encapsulation with a silica shell, especially with higher Fe loading, protected the iron carbide nanoparticles more effectively from sintering and re-oxidation, which are known to be the main causes of deactivation. 

The TEM images and SAED patterns are shown in Figure 8. Figure 8a–c show the morphology of the spent χ-Fe_5_C_2_@SiO_2__L catalyst, and the morphology of the catalyst was well maintained even after the FTS reaction. In addition, as the average crystallite size calculated with the XRD result did not change during the reaction, TEM images showed that the crystallite size of the iron nanoparticles was almost unchanged in the spent catalyst. In Figure 8d, χ-Fe_5_C_2_, θ-Fe_3_C, and α-Fe_2_O_3_ phases can be observed. Figure 8e–g show that the morphology of the catalyst after 24 h of FTS appeared sustained compared with that of the fresh χ-Fe_5_C_2_@SiO_2__H catalyst. The SAED patterns in Figure 8h indicate that both the χ-Fe_5_C_2_ and Fe_3_O_4_ phases were observed. Although sintering and phase transformation did not occur significantly during the exothermic reaction over the χ-Fe_5_C_2_@SiO_2__H catalyst, slight deactivation was observed as shown in Figure 5c. The amorphous silica peak of the spent catalyst in Appendix A shifted slightly to a larger 2θ value, compared with that of the fresh catalyst. The pore structure of the silica shell appeared to undergo thermal contraction because hydroxyls and decomposed molecules of chemical agents from the silica structure were eliminated during the reaction. This might be due to the removal of the calcination step during the preparation of encapsulated catalysts in order to avert phase change or oxidation of active iron carbide. To verify the structural change of the silica shell, the catalyst was characterized by N_2_-physisorption and SAXS analysis. The physisorption result of the spent χ-Fe_5_C_2_@SiO_2__H catalyst is shown in Figure 9a,b. The BET surface area and the pore volume were greatly increased, as shown in Appendix A, which could be the result of the elimination of CTAB and PVP and the condensation of small molecules such as water. The result of SAXS analysis in Figure 9c shows that the q vector for the (100) d-spacing of the ordered mesoporous fraction did not change after the reaction, indicating that the structural change of the ordered mesoporous fraction was slight, and the intensity increased owing to the elimination of CTAB, PVP, and water molecules associated with the silica shell.

Mössbauer emission spectroscopy (MES) was conducted for identifying kinds and fractions of iron phases in χ-Fe_5_C_2_@SiO_2_ catalysts. MES results of fresh and spent χ-Fe_5_C_2_@SiO_2_ catalysts are shown in Figure 10, and the specific parameters are shown in Table 2. The MES spectra of χ-Fe_5_C_2_@SiO_2__L showed two sextets for Fe_3_O_4_, three sextets for χ-Fe_5_C_2_, one sextet for Fe_3_C, and one doublet in the catalyst regardless of evaluating the FTS reaction, as shown in Figure 10a,b. The doublet can be assigned to α-Fe_2_O_3_, considering the XRD spectra shown in Figure 2a and Figure 7a. By comparison of the relative areas of the fresh catalyst with those of the spent χ-Fe_5_C_2_@SiO_2__L catalyst shown in Table 2, it can be seen that χ-Fe_5_C_2_ and Fe_3_C slightly decreased, Fe_3_O_4_ decreased from 46.67% to 34.75%, and α-Fe_2_O_3_ increased from 7.78% to 21.09%.

MES spectra of χ-Fe_5_C_2_@SiO_2__H were found to be quite different from those of χ-Fe_5_C_2_@SiO_2__L. They exhibited only three sextets for χ-Fe_5_C_2_ and one doublet in the catalyst regardless of evaluating the FTS reaction, as shown in Figure 10c,d. The doublet could be assigned to Fe_3_O_4_, considering the XRD spectra shown in Figure 2b and Figure 7b. The Fe_3_O_4_ phase in the χ-Fe_5_C_2_@SiO_2__H catalyst was shown in the form of one doublet instead of two sextets in the MES spectra because the average size of Fe_3_O_4_ was quite small. Other iron species were not observed even though the catalyst went through 24 h of reaction. In the fresh catalyst, relative areas of χ-Fe_5_C_2_ and Fe_3_O_4_ were estimated as 80.55% and 19.45%, respectively. Those were found to be 86.50% and 13.50%, respectively, in the spent catalyst, as shown in Table 2. Only a slight change in the area of the χ-Fe_5_C_2_ phase was observed for the spent catalyst, indicating that the encapsulating silica shell protected iron nanoparticles from re-oxidation and phase transformation to something else. Due to the surrounding restriction by tightly encapsulating silica shell, carbon atoms around the nanoparticle seemed to have difficulty in penetrating into the χ-Fe_5_C_2_ phase of the confined nanoparticle since it seemed very difficult to modify the original crystalline structure into another iron carbide structure with higher carbon content in such restricted surroundings. Furthermore, it seemed that internal carbon atoms of the χ-Fe_5_C_2_ phase had similar difficulty in leaching out. A similar report indicated that the χ-Fe_5_C_2_ phase was unchanged or slightly changed after the FTS reaction when the catalyst was prepared by the formation of iron oxalate dihydrate particles followed by hydrothermal treatment. The spent Fe_5_C_2_@C catalyst showed slight sintering, which was proved by the sharper diffraction peak of the XRD spectrum [38].

## 4. Conclusions

Nanoparticles of the pure χ-Fe_5_C_2_ phase were successfully synthesized using a wet-chemical route, followed by encapsulation with a partially ordered mesoporous silica shell. By doing so, it was shown that the pre-treatment step in FTS reaction could be eliminated. Due to the silica shell’s physical restriction of the nanoparticles, sintering seemed successfully suppressed, which was elucidated by XRD and TEM methods. Structural analysis by small angle X-ray scattering method showed that the protective silica shell had a partially ordered mesoporous structure with a short range. Wide-angle XRD results showed that the crystallite size of the active χ-Fe_5_C_2_ phase did not seem to significantly change during the reaction. Using MES and XRD methods, it was found that the composition and the phase of active χ-Fe_5_C_2_ seemed well maintained during such an exothermic FTS with water produced as a by-product at high temperature and pressure. The Fischer–Tropsch activity of χ-Fe_5_C_2_ in this study was comparable to or higher than those reported in previous studies. In addition, CO_2_ selectivity was found to be very low after stabilization.

## Figures and Tables

**Figure 1 nanomaterials-12-03704-f001:**
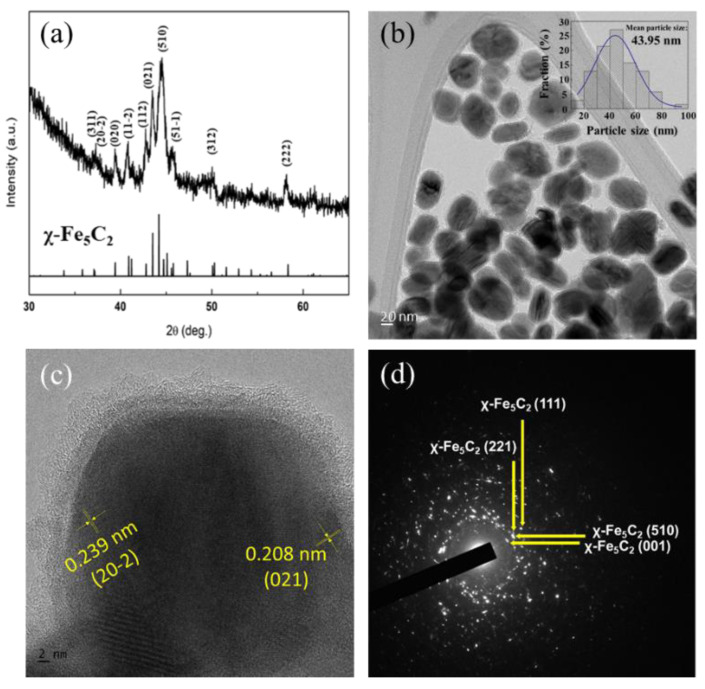
(**a**) XRD pattern; (**b**)TEM image; (**c**) HRTEM image; (**d**) SAED pattern of χ-Fe_5_C_2_ nanoparticles.

**Figure 2 nanomaterials-12-03704-f002:**
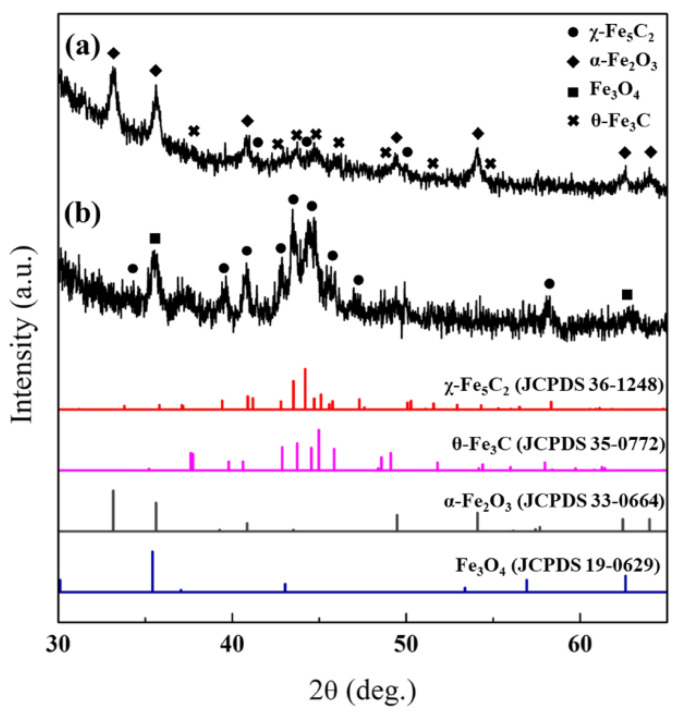
XRD pattern of fresh catalysts: (**a**) χ-Fe_5_C_2_@SiO_2__L; (**b**) χ-Fe_5_C_2_@SiO_2__H.

**Figure 3 nanomaterials-12-03704-f003:**
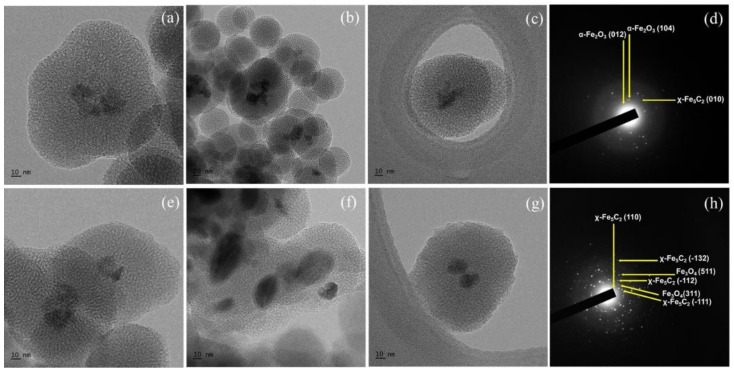
(**a**–**c**) TEM images and (**d**) SAED pattern of fresh χ-Fe_5_C_2_@SiO_2__L catalyst; (**e**–**g**) TEM images and (**h**) SAED pattern of fresh χ-Fe_5_C_2_@SiO_2__H catalyst.

**Figure 4 nanomaterials-12-03704-f004:**
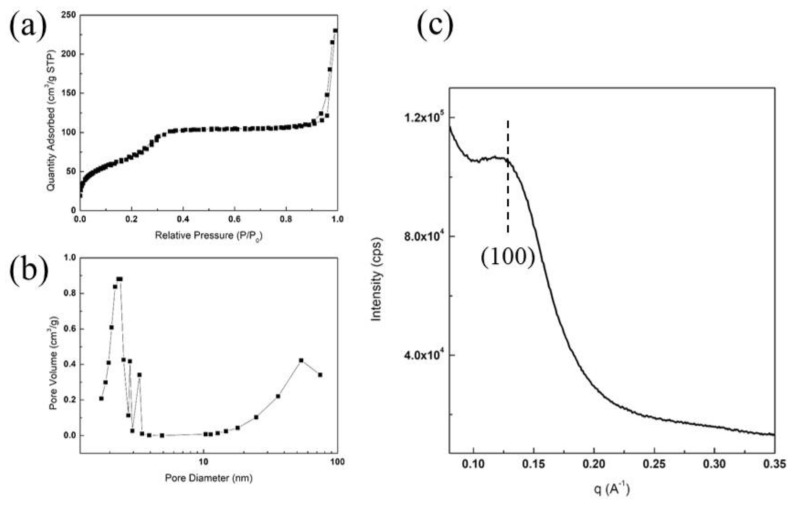
N_2_-physisorption and SAXS results: (**a**) N_2_-adsorption isotherm and (**b**) pore size distribution curve; (**c**) small-angle X-ray scattering pattern of fresh χ-Fe_5_C_2_@SiO_2__H catalyst.

**Figure 5 nanomaterials-12-03704-f005:**
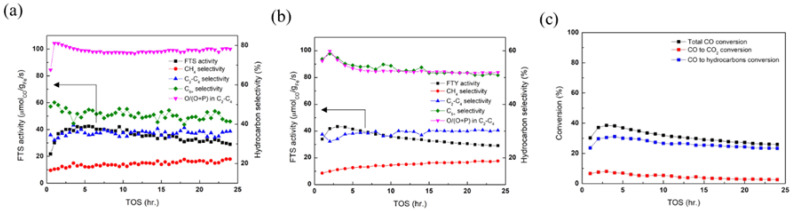
(**a**) The FTS reaction activity and hydrocarbon selectivity over the χ-Fe_5_C_2_@SiO_2__L catalyst during the 24 h time on stream; (**b**) the FTS reaction activity and hydrocarbon selectivity over the χ-Fe_5_C_2_@SiO_2__H catalyst during the 24 h time on stream; (**c**) CO conversions to hydrocarbons and CO_2_ of the χ-Fe_5_C_2_@SiO_2__H catalyst, the total CO conversion is the sum of the CO to hydrocarbon conversion and the CO to CO_2_ conversion. Reaction conditions: 0.3 g catalyst, 300 °C, 10 bar, H_2_/CO = 2.1, GHSV = 4000 mL g_cat_^−1^·h^−1^.

**Figure 6 nanomaterials-12-03704-f006:**
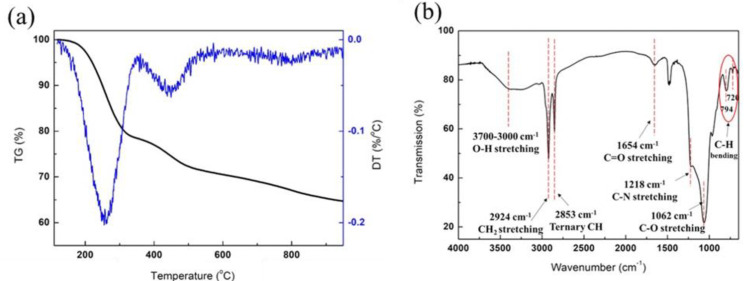
TG/DTA and FT-IR results: (**a**) TG/DTA result and (**b**) FTIR spectra of fresh χ−Fe_5_C_2_@SiO_2__H catalyst.

**Figure 7 nanomaterials-12-03704-f007:**
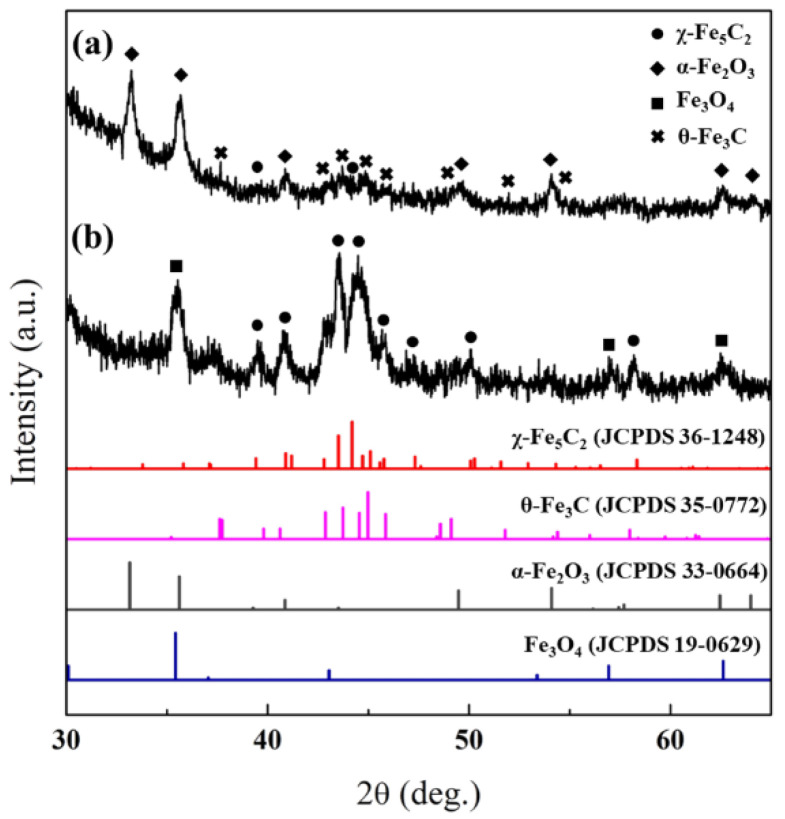
XRD pattern of spent catalysts after reaction: (**a**) χ-Fe_5_C_2_@SiO_2__L; (**b**) χ-Fe_5_C_2_@SiO_2__H.

**Figure 8 nanomaterials-12-03704-f008:**
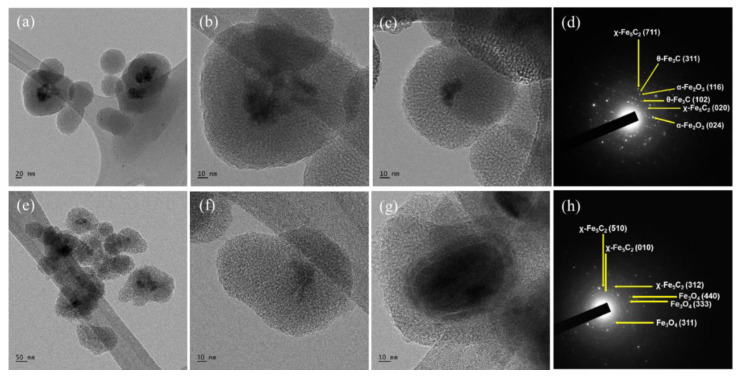
(**a**–**c**) TEM images and (**d**) SAED pattern of spent χ-Fe_5_C_2_@SiO_2__L catalyst; and (**e**–**g**) TEM images and (**h**) SAED pattern of spent χ-Fe_5_C_2_@SiO_2__H catalyst.

**Figure 9 nanomaterials-12-03704-f009:**
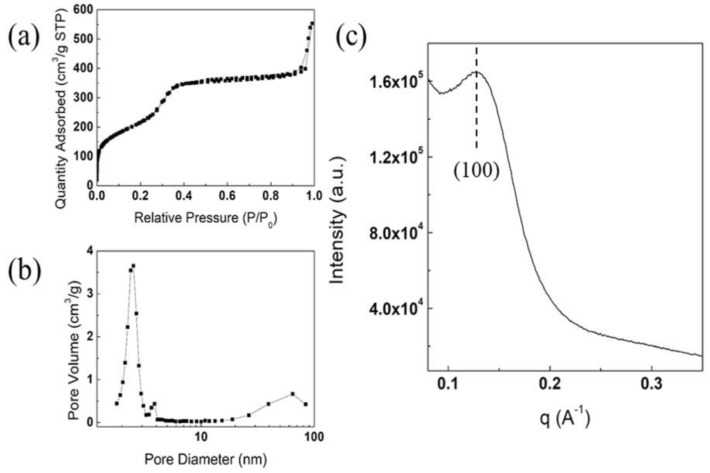
N_2_-physisorption and SAXS results: (**a**) N_2_-adsorption isotherm and (**b**) pore size distribution curve; (**c**) small-angle X-ray scattering pattern of spent χ-Fe_5_C_2_@SiO_2__H catalyst after reaction.

**Figure 10 nanomaterials-12-03704-f010:**
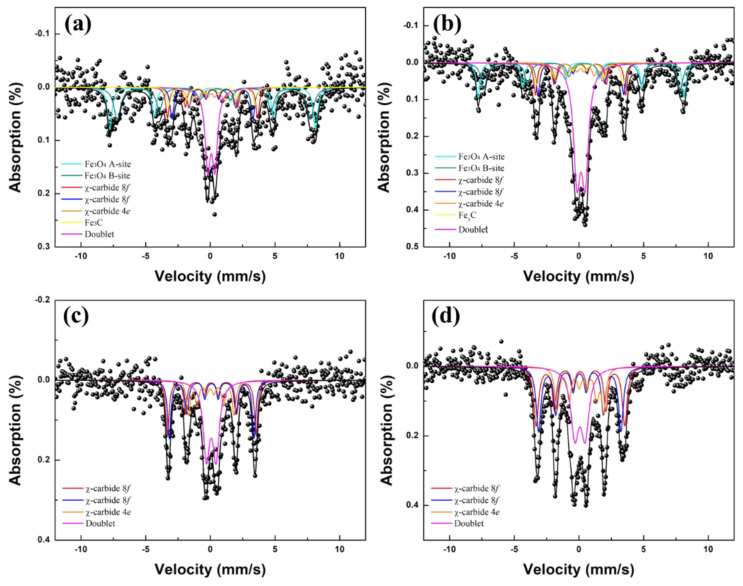
MES spectra of (**a**) fresh χ−Fe_5_C_2_@SiO_2__L catalyst, (**b**) fresh χ−Fe_5_C_2_@SiO_2__L catalyst, (**c**) fresh χ−Fe_5_C_2_@SiO_2__H catalyst, and (**d**) spent χ−Fe_5_C_2_@SiO_2__H catalyst.

**Table 1 nanomaterials-12-03704-t001:** CO conversion, CO_2_ and hydrocarbon selectivity, and FTS activity of χ-Fe_5_C_2_@SiO_2__L, χ-Fe_5_C_2_@SiO_2__H, and other reference catalysts.

Catalyst	Fe Loading (wt.%)	CO conv. (%)	Hydrocarbon Selectivity (%)				CO_2_ sel. (%)	FTS Activity (umol_CO_/g_Fe_/s)	Reaction Condition	Ref.
			CH_4_	C_2_–C_4_	C_5+_	O/(O + P) in C_2_–C_4_				
χ-Fe_5_C_2_@SiO_2__L	3.5	10.8	18.2	34.5	47.3	77.2	-	42.5	300 °C, 10 bar, H_2_/CO = 2.1, GHSV = 4000 mL∙g_cat_^−1^∙h^−1^	This work
χ-Fe_5_C_2_@SiO_2__H	12.2	38.5	15.6	27.1	57.3	56.6	20.9	44.1	300 °C, 10 bar, H_2_/CO = 2.1, GHSV = 4000 mL∙g_cat_^−1^∙h^−1^	This work
FeSi-syn	50.9	22.3	12.9	17.1	70.0	47.6	2.8	7.2	260 °C, 30 bar, H_2_/CO = 2 GHSV = 4000 h^−1^	[13]
15.7Fe/AC	15.7	29.4	18.4	51.1	30.6	44.1	30.1	34.8	260 °C, 20.7 bar, H_2_/CO = 0.9 GHSV = 3000 mL∙g_cat_^−1^∙h^−1^	[29]
Fe25Si	58.9	46.7	15.6	39.2	45.2	33.3	N.A.*	6.7	280 °C, 15 bar, H_2_/CO = 2 GHSV = 2000 mL∙g_cat_∙h^−1^	[30]
CAT-H_2_O	54.5	46.5	11.7	25.9	62.4	N.A.*	30.0	36.0	280 °C, 20 bar, H_2_/CO = 1 GHSV = 12000 mL∙g_cat_^−1^∙h^−1^	[31]
SiO_2_/α-Fe_2_O_3_	35.0	44.9	18.7	47.8	33.5	25.1	25.1	54.0	280 °C, 10 bar, H_2_/CO = 1 GHSV = 5000 mL∙g_cat_^−1^∙h^−1^	[32]
Al_2_O_3_/α-Fe_2_O_3_	35.0	61.6	16.0	47.3	36.7	33.9	33.9	74.1	280 °C, 10 bar, H_2_/CO = 1 GHSV = 5000 mL∙g_cat_^−1^∙h^−1^	[32]

* N.A.: not available.

**Table 2 nanomaterials-12-03704-t002:** Mössbauer parameters of χ-Fe_5_C_2_@SiO_2__L and χ-Fe_5_C_2_@SiO_2__H catalysts.

Sample		δ * (mm/s)	ΔE_Q_ (mm/s)	H_hf_ (kOe)	phase	Area (%)	Composition (%)
χ-Fe_5_C_2_@SiO_2__L	Fresh	0.22	−0.04	493.8	Fe_3_O_4_ (A)	21.39	46.67
		0.36	−0.04	463.1	Fe_3_O_4_ (B)	25.28	
		0.13	0.04	216.3	χ-Fe_5_C_2_ 8f	11.37	34.21
		0.16	0.0	190.4	χ-Fe_5_C_2_ 8f	13.22	
		0.09	0.04	109.4	χ-Fe_5_C_2_ 4e	9.62	
		0.11	0.0	205.9	Fe_3_C	11.34	11.34
		0.08	0.55	-	Doublet	7.78	7.78
	Spent	0.22	−0.04	494.6	Fe_3_O_4_ (A)	15.94	34.75
		0.33	−0.05	470.2	Fe_3_O_4_ (B)	18.81	
		0.10	0.02	216.6	χ-Fe_5_C_2_ 8f	11.72	33.50
		0.11	0.01	202.8	χ-Fe_5_C_2_ 8f	13.07	
		0.09	0.04	112.3	χ-Fe_5_C_2_ 4e	8.71	
		0.10	0.02	207.1	Fe_3_C	10.66	10.66
		0.15	0.63	-	Doublet	21.09	21.09
χ-Fe_5_C_2_@SiO_2__H	Fresh	0.08	0.00	214.4	χ-Fe_5_C_2_ 8f	28.13	80.55
		0.09	0.0	195.3	χ-Fe_5_C_2_ 8f	31.10	
		0.08	0.04	110.9	χ-Fe_5_C_2_ 4e	21.32	
		0.08	0.74	-	Doublet	19.45	19.45
	Spent	0.10	0.01	213.6	χ-Fe_5_C_2_ 8f	30.09	86.50
		0.08	0.0	197.7	χ-Fe_5_C_2_ 8f	33.62	
		0.17	−0.04	113.0	χ-Fe_5_C_2_ 4e	22.79	
		0.08	0.78	-	Doublet	13.50	13.50

* δ: isomer shift, ΔE_Q_: quadrupole shift, Hhf: hyperfine magnetic field.

## Data Availability

The Appendix A above is available at the *Nanomaterials* website.

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
