# Peer review of "Effects of Silica Shell Encapsulated Nanocrystals on Active χ-Fe_5_C_2_ Phase and Fischer–Tropsch Synthesis"

_nanomaterials, 2022, doi:10.3390/nano12203704_

Round 1

Reviewer 1 Report

This manuscript reports the synthesis of encapsulated iron carbide phases, χ-Fe5C2, with mesoporous silica and their use as catalysts for Fischer-Tropsch synthesis. The experimental work is very well structured, although the experimental section is not clear in my opinion, and the characterization techniques are adequate. The spent catalysts are characterized to demonstrated the role of protective silica shell to avoid catalyst sintering and oxidation. Mössbauer, XRD, XPS, FT-IR and N2 physisorption characterization techniques are used. Furthermore, the results are compared with similar studies on this topic. The experimental work seems complete and the discussion are well supported.
At current stage, I recommend the manuscript to be accepted after the following revision:

  • Line 74: Purity of all reagents should be given.
  • Lilne 82: The synthesis of iron carbide nanoparticles and χ-Fe5C2@SiO2 catalysts should be reproducible and is not clear enough. Thus, authors should be more precise in the description. For i.e.,

-During the synthesis of iron carbide nanoparticles, is the mixture of reactants under inert atmosphere all the procedure?. Which amount of hexane is used to “wash” the iron carbide nanoparticles?. Which temperature and time of washing? It is subsequently washed with other solvent?

-       Line 90:“…washed with magnetic separation”. This is not clear.

-       Line 93: “Predetermined mass of χ-Fe5C2 nanoparticles were mixed with 3 g of polyvinylpyr-93 rolidone (PVP)”. “Predetermined” is unprecise.

-       Line 96. Which solution was “evaporated” in the rotatory evaporator. Solution A or Solution B?

-       Line 108: Also, the contact time of PVP-protected χ-Fe5C2 nanoparticles with aqueous solution was controlled to protect from oxidation. Except for these, the synthesis proceeded in the same manner. Authors should be more precise to guarantee catalysts preparation reproducibility.

  • Particle size was estimated by XRD using Scherrer´s equation. Which is the value for K used by the authors?
  • Lines 226, 229, and Table S1: BET surface values should be given without decimals, due to the accuracy of the BET method. The same for the pore volume, one decimal should be enough in this case.
  • Figure 5a and Figure 5b are not comparable due to the different iron loading presented in χ-Fe5C2@SiO2_L (3.5 wt. %) and χ-Fe5C2@SiO2_H (12.2 wt%). Instead of CO conversion, Iron time yield expressed as µmolCO/gFe/s should be represented for these two catalysts for comparative purposes.
  • Table 1 should show the FTS reaction conditions (temperature, pressure, H2/CO ratio and GHSV (mL/gFe/h) used in each reported work to have a better vision of the outstanding behavior of the catalysts used in the present work.
  • Line 423: Authors suggests that the active sites confinement could prevent the deposition of a higher carbon content. Did the authors measured TGA analysis of the spent catalysts to confirm this? Carbon deposition is one of the main reasons of iron-based catalysts deactivation in FTS.

Reviewer 2 Report

It is an interesting paper on the synthesis of encapsulated nanoctrystals of the Fe5C2 phase in a silica shell by the wet-chemical route. The synthesized samples with different amount of active phase were tested in Fisher-Tropsch synthesis. The authors investigated the synthesized catalysts by nitrogen adsorption/desorption, XRD, SAXS, TEM, XPS, ICP-OES, TGA, FT-IR and Mössbauer spectroscopy. It is a meaningful work. Nevertheless, I suggest making some major corrections before its publication in Nanomaterials:

1)     I suppose that the changes in N2 sorption results of catalysts before and after the FTS synthesis are not due to the decomposition of CTAB and PVP. The N2 sorption was performed after degassed of catalysts at 350˚C. Thus, in my opinion, the fresh catalysts did not contain the CTAB which was removed during this pretreatment step (the decomposition of CTAB occurred below 350 ˚C – TGA analysis). The PVP starts to decompose at approximately 330 ˚C (TGA analysis). However the pretreatment of catalysts before the N2 sorption was performed in vacuum which could decrease the temperature of PVP decomposition. Thus, I kindly recommend performing the elemental analysis to determine the real concentrations of CTAB and PVP (nitrogen content from elemental analysis) in the samples before and after FTS synthesis. The elemental analysis allow to detect the concentration of both substances in whole sample. Such data will give an answer if all CTAB and PVP were removed from the catalysts and caused the changes in morphology of catalysts.

2)     The degass step of N2 sorption could be done at a lower temperature, that is, 120 ˚C but with the elongation of the time. At this temperature, the CTAB and PVP should not be removed from the catalysts.

3)     In Figure S2, the XRD patterns of fresh and spent catalysts were not marked. Please add the more detailed description into this Figure (i.e. a – spent catalyst, b – fresh catalyst). The description of Figure S2 is given in the main text after Figure S3.

4)     I kindly recommend performing the deconvolution of the XPS spectra shown in Figure S3. The position of the described peaks will be better visible and thus will be possible to compare. What spectra are related to spent and fresh catalysts? Please add the more detailed description of figure.

5)     The temperature of N2 physisorption was given in Kelvin not in Celsius.

Round 2

Reviewer 2 Report

The authors corrected the manuscript according to the Reviewers corrections which improved the paper. Thus, I recommend it for publication in Nanomaterials journal.